# Ionic Strength Dependence of the Complex Coacervation between Lactoferrin and β-Lactoglobulin

**DOI:** 10.3390/foods12051040

**Published:** 2023-03-01

**Authors:** Rima Soussi Hachfi, Pascaline Hamon, Florence Rousseau, Marie-Hélène Famelart, Saïd Bouhallab

**Affiliations:** INRAE, Institut Agro, STLO, 65 Rue de Saint Brieuc, F-35042 Rennes, France

**Keywords:** complex coacervation, ionic strength, proteins, desalting

## Abstract

Heteroprotein complex coacervation is an assembly formed by oppositely charged proteins in aqueous solution that leads to liquid–liquid phase separation. The ability of lactoferrin and β-lactoglobulin to form complex coacervates at pH 5.5 under optimal protein stoichiometry has been studied in a previous work. The goal of the current study is to determine the influence of ionic strength on the complex coacervation between these two proteins using direct mixing and desalting protocols. The initial interaction between lactoferrin and β-lactoglobulin and subsequent coacervation process were highly sensitive to the ionic strength. No microscopic phase separation was observed beyond a salt concentration of 20 mM. The coacervate yield decreased drastically with increasing added NaCl from 0 to 60 mM. The charge-screening effect induced by increasing the ionic strength is attributed to a decrease of interaction between the two oppositely charged proteins throughout a decrease in Debye length. Interestingly, as shown by isothermal titration calorimetry, a small concentration of NaCl around 2.5 mM promoted the binding energy between the two proteins. These results shed new light on the electrostatically driven mechanism governing the complex coacervation in heteroprotein systems.

## 1. Introduction

Oppositely charged polymers interacting mainly through electrostatic interactions can undergo a spontaneous liquid−liquid phase separation (LLPS) into polymer-rich dense phase called coacervates and a less concentrated phase called the dilute phase. Complex coacervation, known as LLPS, has been and still is a subject of intense experimental and theoretical interest since the pioneer research of Bungenberg De Jong and Kruyt one century ago [1]. Although several theoretical models have been provided to describe complex coacervation, none of them were able to perfectly explain it [2,3,4,5,6]. Several studies, based on those models, described complex coacervation phenomenon as a four-step process. The first step is the spontaneous formation of both symmetrical and random heterocomplexes by electrostatic attraction (building blocks). The building blocks, also called primary units, come together to form soluble complexes. The third step involves the rearrangement of soluble complexes into spherical micrometric droplets, characteristic of complex coacervation [7]. Finally, the droplets coalesce, thus forming the dense phase of the coacervates [8]. Interest in complex coacervation is remarkably increasing as it is an ambitious undertaking that will open enormous opportunities for numerous applications in the food, cosmetics, pharmaceutical and biological fields [9,10]. Moreover, complex coacervation can occur across a wide-variety of charged macromolecules, such as protein–polysaccharide, protein–synthetic polyelectrolyte, polyelectrolyte–polyelectrolyte and protein–proteins mixture. However, heteroprotein complex coacervation (HPCC), i.e., involving two or more proteins, is comparatively understudied [11,12,13,14]. Part of the published works on heteroprotein complex coacervation focused on the assembly between lactoferrin (LF) as basic protein and β-lactoglobulin (βLG) as acidic protein. In these studies, multiscale characterization of the LF–βLG formed coacervates were conducted using biophysical tools such as Small Angle X ray Scattering (SAXS) measurements [15] or solid-state Nuclear Magnetic Resonance NMR [16]. The potential applications of LF–βLG coacervates in food and their effectiveness for encapsulation of bioactives were also reported [17]. In these studies, LF–βLG coacervates showed a high vitamin encapsulation yield, around 98% [17]. In addition to that, thanks to their high viscosity, the coacervates can be used as food texturizing agents that offer a good substitute for exogenous additives, such as polysaccharides, that will help in developing “clean label” functional food products [18]. For HPCC, the most subtle is to set the physico-chemical parameters (pH, stoichiometry, concentration, etc.) for optimal coacervation. Yan et al. [19] reported that LF–βLG coacervation is favored under a narrow range of pH 5.7 to 6.2, with lower ionic strength value and a total protein concentration between 10 and 40 g/L. For their part, Anema & de Kruif [20], noticed that LF–βLG coacervates were formed in the pH region of 5.0–7.3 and ionic strength lower than 100 mM. In a previous work, the pH range has been narrowed and optimal LF–βLG coacervation occurred at pH 5.5 [21]. Furthermore, the coacervates yield has been compared when mixing LF with the two isoforms A and B of βLG. Interestingly, the higher coacervates yield was recovered with the most acidic isoform, i.e., βLG isoform A (one more aspartic acid residue). These results underline the high sensitivity of LF–βLG coacervation to a small variation in the net protein charge [12,21]. Behind the pH, the ionic strength is another important parameter that controls the net protein charge and consequently the electrostatically driven heteroprotein complex coacervation. Therefore, the aim of this present work is to gain more insight on the effect of the presence of salt on the interaction and complex coacervation between LF and βLG.

## 2. Materials and Methods

LF with a purity of 90 g/100 g and iron saturation of 10–20 mol iron–mol protein according to technical specification was purchased from Fonterra Cooperative Group (Auckland, New Zealand) and used without further purification. Commercial bovine βLG, containing both A and B variants, was further purified before use; as βLG is prompt to self-aggregation during long storage, the non-native and aggregated species were regularly removed by dispersing in ultrapure water (30 g/L), adjusting to pH 5.2 with 1 M HCl and then storing at 30 °C for 10 min to precipitate aggregated and non-native forms. The dispersion was then centrifuged at 36,000× *g* for 10 min at 25 °C (Avanti, J-26S XP BioSafe Three-Phase Non-IVD Centrifuge, Beckman Coulter, Villepinte, France). The supernatant containing native βLG was adjusted to pH 7.0 with 1 M NaOH, freeze-dried and stored at −20 °C until later. The protein purity in obtained powder was around 95% as assessed by HPLC analysis. Sodium chloride (NaCl) was purchased from VWR Chemicals (Rosny-sous-Bois, France). (2-(N-morpholino) ethanesulfonic acid hydrate (MES) was purchased from Sigma-Aldrich (St. Louis, MO, USA) and all other chemicals were of analytical grade.

### 2.1. Preparation of Samples and Direct Mixing

MES buffer (10 mM) was used as solvent in all experiments and was prepared by solubilizing MES powder in ultra-pure water. Solid NaCl was added to reach targeted concentration and then adjusted to pH 5.5 using 1 mM NaOH solution.

LF and βLG Protein powders were solubilized in MES buffer with desired concentration of NaCl (0–400 mM), and the pH was readjusted to 5.5 when required. This pH value was found to be optimal for complex coacervation between the two whey proteins at the current stoichiometry and total protein concentration [17]. In fact, this pH value lies between the two isoelectric points of the proteins; 5.2 and 8.8 for βLG and LF, respectively [12]. The exact protein concentrations were determined in the two stock solutions by absorbance at 280 nm with a SAFAS UV MC2 spectrophotometer (SAFAS UV MC2, Safas, Monte-Carlo, Monaco) using 0.96 L/g.cm and 1.47 L/g.cm as extinction coefficients for βLG and LF, respectively.

For direct mixing experiments, conducted in duplicate, solutions of βLG and LF prepared in 10 mM MES buffer at various NaCl concentrations were mixed at room temperature to reach a final concentration of 0.5 mM and 0.05 mM for βLG and LF, respectively, which means a molar ratio of 10/1. The direct mixing experiments were conducted at least in duplicate, and the mean and the standard deviation are plotted

### 2.2. Complex Coacervates by Desalting

The formation of complex coacervates between the two proteins prepared at various salt concentrations (0, 100, 200 and 400 mM NaCl) was monitored during continuous desalting against 10 mM MES buffer, pH 5.5 using dialysis membranes. Ten mL of each mixture were put into a 6–8 KDa molecular weight cut-off (MWCO) dialysis membrane (diameter = 14.6 mm) (SpectraPor, Repligen Corporation, CA, USA). The dialysis membrane was then submerged into a dialysis bath containing 1 L of MES buffer under constant stirring at room temperature. Fourteen aliquots were taken from the dialysis bag at different times during the 24 h of dialysis for analyses. Dialysis experiments of the 10 mM MES buffer with studied NaCl concentrations and without proteins were also performed as dialysis control experiments. Successive low volume sampling does not affect subsequent measurements

During dialysis experiments, conductivity of the dialysis bath was measured using an electrical conductivity meter (HI98192, Hanna instrument, Strasbourg, France). The probe of the conductivity meter (HI763133, Hanna instrument, Strasbourg, France) was submerged into the bath throughout the dialysis time to make sure that the ion exchange between the dialysis bag and the bath is taking place. Different samples were collected from inside the bag, and the salt concentration was measured using an electrical conductivity meter (HI98192, Hanna instrument, Strasbourg, France). The dialysis experiments were conducted at least in duplicate.

### 2.3. Turbidity Measurements

The turbidity caused by the spontaneous formation of coacervates as spherical droplets in LF–βLG direct mixtures and during the dialysis experiments was measured at 600 nm using a microplate spectrophotometer (Multiskan™ GO, Fisher Scientific, Strasbourg, France). In fact, the spectrophotometer provides absorbance measurements that could be converted to turbidity using the following equation:(1)$$Turbidity\;\left({\mathrm{cm}}^{-1}\right)=\frac{2.303\times A}{L}$$
where *A* is absorbance at 600 nm and *L* (cm) is the light path length corresponding to the height of the liquid column into the microplate well [22].

### 2.4. Coacervate Yield

Protein partition was determined by protein quantification after phase separation. Dilute and coacervate phases were separated by centrifugation (Heraeus Biofuge Primo, Thermo Scientific, Waltham, MA, USA) at 28,000× *g* for 30 min. Protein content in the supernatant was quantified by liquid chromatography (UltiMate 3000 HPLC, Thermo Fisher Scientific, Waltham, MA, USA) as previously described [21]. A PLRP-S column (300 Å, 2.1 × 150 mm, 8 µm) with a flow rate of 0.2 mL/min (Agilent Technologies, Santa Clara, CA, USA) was used. Milli-Q water containing 1.06 ‰ (*v*/*v*) of trifluoroacetic acid and an 80/20 acetonitrile/milli-Q water (*v*/*v*) mixture containing 1.0 ‰ (*v*/*v*) of trifluoroacetic acid were used for elution. The absorbance at 280 nm was measured during the elution using a Waters 2487 detector.

The proteins content in the coacervate phase was calculated by subtracting the proteins content in the supernatant from the total initial protein concentration.

The coacervate yield (protein yield in the coacervate phase) was calculated using the following equation:(2)$$\mathrm{The}\;\mathrm{coacervate}\;\mathrm{yield}\;\left(\%\right)=\frac{\mathrm{final}\;\mathrm{protein}\;\mathrm{concentration}\;\mathrm{in}\;\mathrm{the}\;\mathrm{coacervates}}{\mathrm{Initial}\;\mathrm{protein}\;\mathrm{concentration}}\times 100$$

### 2.5. Phase Contrast Microscopy

Optical microscopy was used to check that coacervates rather than amorphous aggregates were formed during direct mixing and dialysis experiments. Observations were conducted at room temperature using an Olympus phase contrast microscope (BX51TF, Rungis, France) set at the magnifications 100×.

### 2.6. Isothermal Titration Calorimetry (ITC)

The ITC experiment was performed at two temperatures, 25 °C and 35 °C, using a VP-ITC micro-calorimeter (MicroCal VP-ITC, Malvern Panalytical, Malvern, UK) by successive injections of LF solution (0.25 mM) into a βLG solution (0.1 mM) loaded in sample cell (1.425 mL). Titration experiments were performed at different NaCl concentrations between 0 and 20 mM. All solutions, prepared in 10 mM MES buffer pH 5.5, were degassed under vacuum before titration experiments. The reference cell was filled with NaCl solutions in the respective concentration for each measurement, and the sample cell was filled with βLG solution. The LF solution in the syringe was also set at the same NaCl concentration. βLG was titrated with 25 successive injections of 10 µL of LF solution. The initial delay was set at 60 s, and the stirring rate inside the sample cell was set at 300 rpm to ensure the homogeneity of the cell solution during titration. The interval between injections was 200 s to reach the thermodynamic equilibrium. For each ITC experiment, a reference titration was performed by titrating LF solutions directly into 10 mM MES buffer containing the studied NaCl concentration. A negligible signal was associated with this reference injection, which was subtracted from the corresponding experimental signal. The ITC data were fitted using graphical user interface (GUI) of PyTC (version 1.2.2, University of Oregon, OR, USA), an open-source python software [23]. The ITC experiments were performed at least in duplicate.

### 2.7. Statistical Analysis

The effect of salt concentration on coacervation yield and turbidity during direct mixing and dialysis experiments and on titration by ITC was tested using analyses of variance (ANOVA) with one factor. When the ANOVA was significant (*p* < 0.05), Tukey’s multiple comparison test was used for paired comparisons of means with *p* < 0.05. Tests were performed using R Studio, a software package developed by a community (version 2021.09.1, Boston, MA, USA).

## 3. Results and Discussion

### 3.1. Interactions between βLG and LF

#### 3.1.1. Direct Mixing Experiments

The effect of ionic strength on the interaction–complex coacervation was studied by preparing individual proteins in the required salt concentration before mixing. Figure 1 shows the turbidity and the coacervate yield at different ionic strengths from 0 mM to 100 mM. Turbidity and coacervate yield were maximal without added salt and decreased with increased ionic strengths to reach a value close to zero at 20 mM for both the turbidity and the coacervates yield. At this concentration, no LLPS was detected.

These results are in agreement with those reported by Yan et al. [19] for the same coacervates system at a close pH value, say pH 5.9. In their study, the yield in βLG and LF has been monitored as a function of NaCl concentrations (0−100 mM) using size exclusion chromatography. Both proteins showed a decrease in their yield with the increase of the ionic strength. A concentration of 20 mM of NaCl was also found to be a critical salt concentration for coacervation. Moreover, in the same paper, the authors noticed that at this salt concentration, and after centrifugation, a white precipitate of βLG aggregates was observed instead of LF–βLG coacervates. The effect of NaCl on the heteroprotein coacervates was also conducted on another protein system, β-casein–LF [19]. Similar ionic strength dependency of complex coacervation between these two proteins at pH value of 6.5 was reported. However, for β-casein–LF, complex coacervation was still observed even for ionic strength higher than 140 mM, a value 7 times higher than that found for LF–βLG. Hence, the concentration of NaCl tolerated by LF–βLG system is lower than that tolerated by β-casein–LF system. This difference might be explained by the random coil structure of β-casein as intrinsically disordered protein with high hydrophobicity compared to βLG [24]. Consequently, the subtle sensitivity to salt is completely protein structure dependent.

The coacervate yield decreased drastically with increasing ionic strength (Figure 1). The yield of whey protein and gum arabic complex coacervates at pH 4 decreased when increasing NaCl concentration and was completely suppressed at ionic strength higher than 60 mM [25]. Hence, the value found for protein–polysaccharide was higher than that found in the present work for heteroprotein LF–βLG system. In addition to that, the same authors reported that the kinetics of the phase separation slows down upon addition of salt because the coalescence of the coacervate droplets takes more time and water being slowly released from the coacervate phase.

Even though the decrease in the coacervate yield is a great proof of the effect of ionic strength on the formation of coacervates, turbidity measurement is also a powerful indicator of the coacervation. A slight increase of NaCl concentration induces a rapid decrease of turbidity (Figure 1). As checked by microscopic observations, addition of salt decreased both the size and the number of formed droplets; although further and systematic image analyzes are required, it seems that the droplet diameter decreased significantly after addition of NaCl. This behavior is not specific to HPCC since it was generally observed for several systems, such as polysaccharide–polysaccharide and polysaccharide–protein [26,27,28,29]. Such evolution of the turbidity as a function of the increase in the ionic strength highlights the predominant role of attractive electrostatic forces in the complex coacervation process. Hence, stronger attractive interactions between biopolymers lead to a more turbid solution [30]. The major difference lies in the salt sensitivity threshold, which is strongly dependent on the structures of the mixed macromolecules.

This huge dependency of LF–βLG complex coacervation on ionic strength obeys the mechanism explained a long time ago by Bungenberg de Jong [1]: the presence of microions screens the charges of the polymers, which weakened attractive forces between them and disrupted the intermolecular electrostatic interactions. As complex coacervation is an electrostatically driven process, the weakening of the electrostatic interaction hampers the complex formation and hinders the occurrence of LLPS. As a matter of fact, the theory of Overbeek and Voorn [2] elucidates that, before reaching a critical salt concentration, the polymer mixture was able to get separated into a polymer-rich and a polymers-poor phase (the LLPS). However, with the addition of microions, the composition of the two phases became very close until reaching a critical point, beyond which LLPS is abolished [2]. Overall, A high sensitivity of LF–βLG complex coacervation to salt with a critical NaCl concentration of 20 mM above which the coacervation can no longer occur was reported. Addition of increasing salt concentration (from 0 to 20 mM) decreased the overall net charge of the two proteins, as the measured zeta potential decreased from −14 to −12 mV for βLG and from +12 to +4 mM for LF (results not shown). Consequently, increasing NaCl concentration affected the protein surface net charge, shifted the protein’s isoelectric points and reduced the long-range electrostatic attractions.

#### 3.1.2. LF–βLG Interaction Energy

ITC experiments were conducted to provide a detailed thermodynamic description on how low ionic strength affects the interaction and association between βLG and LF. The heat flow versus time (raw data) profiles associated with the titrations of LF into βLG at different NaCl concentration and at 25 °C are shown in the top panels of Figure 2. The actual heat associated with the interactions (Figure 2 bottom panels) was obtained by integrating the peaks of the top panels and subsequently subtracting the heat produced from the titration of LF into the buffer. Based on the resulting titration profiles, the heat flow was negative (ΔH < 0), meaning that the sum of interactions and other phenomena taking place is an exothermic process. The presence of NaCl affects signal intensity but did not change the negative signature, suggesting that the nature of non-Coulombic interactions (such as hydrophobic interactions and hydrogen bonding) was not altered by the salt-shielding effect. Exothermic processes during the interaction between two oppositely charged macromolecules have been reported for various other systems [21,31,32,33,34,35]. The exothermic nature of an enthalpically driven complexation process is generally attributed to the predominance of electrostatic interactions. During the titration at very low added salt, a higher response in heat change was observed during the first injections, which gradually decreased over the titration and tended to saturation with increased protein molar ratio.

The thermograms at very low salt concentration (Figure 2A–C) show two inflexion points that could describe two successive steps, which means that two possible events came into play during the complexation of βLG with LF. The second inflexion point (above molar ration of 0.2) disappeared with higher ionic strength values (Figure 2D,E). Since increasing the ionic strength can enhance the hydrophobic interaction [36], the second event might be hampered by the reinforcement of the hydrophobic effects. To confirm this hypothesis, the same ITC experiment was conducted without added salt, but at 35 °C as a temperature increment promotes hydrophobic effects. As illustrated in Figure 3, when temperature increased from 25 to 35 °C without added salt, the second signal was lost. Hence, hydrophobic effects might prevent the second mechanism that normally took place at a high molecular ratio. Given that an increase in temperature or ionic strength enhance the hydrophobic effects causing protein aggregation [37], a possible explanation could be that the interaction–coacervation between βLG and LF is a question of predominance between electrostatic interactions and hydrophobic effects. A decrease in electrostatic attractive forces being predominant over the effect of an increase in hydrophobic effects.

The formation of coacervates, according to a two-step process as detected by ITC, has been described for other macromolecular systems [32,38,39]. The first enthalpy-driven step is attributed to electrostatic interactions or ion pairing and leads to the formation of soluble complexes. The second step, rather entropy-driven, is attributed to the self-aggregation of these complexes into coacervates. This explanation fits well for the studied LF–βLG system since the second inflexion point was suppressed with increasing ionic strength, concomitantly to the disappearance of LLPS as shown by turbidity measurements presented above. The second event can thus be assigned to the complex coacervation step between the two proteins.

The explanation of what happens during the two steps is not easy since, for macromolecular interactions, each thermodynamic signal is a result of the contribution of several phenomena: interaction, protein conformational change, release of water, protons and other ions, complexation, reorganizations, aggregation, etc. The overall measured signal therefore comes from endothermic and exothermic reactions whose final absolute value is the result of the dominant energy. To go further in the exploration of the thermodynamic changes occurring during titration, the binding isotherms were fitted using PyTC to determine the enthalpy and associated binding constant (Table 1). Without added salt, the interaction exhibited high enthalpy change and high affinity constant K_a_ in the micromolar range. Interestingly, the addition of 2.5 mM NaCl further promotes the interaction with a +25% gain in enthalpy value and twofold increase of the affinity constant. Hence, a small amount of added salt favors the interactions between the two oppositely charged proteins. For higher added NaCl concentrations, the screening effect of salt on the interaction and association between the two proteins started to be observed as reflected in the ITC signals and in the significant decrease of K_a_ and ∆H values (Table 1).

Burgess [40] reported that the general trend of gelatin–acacia coacervate yield increased with an increase in ionic strength up to a maximum and then decreased with a further increase in ionic strength. This ‘‘salting-in like’’ trend was explained as a consequence of the effect of added salt on the extent of coiling and charge densities of the involved macromolecules. The results found here for LF and βLG are consistent with such reported data except that the concentration of NaCl tolerated by LF–βLG (HPCC system) is much less than that tolerated by gelatin–acacia coacervates. In general and compared to HPCC, higher concentration of salt is needed to screen the interaction–complexation in protein–polysaccharide systems, and a relatively small amount can indeed promote the complexation as confirmed recently for pea protein–chitosan and ovalbumin–carboxymethylcellulose [41,42].

The enthalpy change reflects the contribution of hydrogen bonds, electrostatic and van der Waals interactions [34], and its decrease is hence expected with screening effect at increasing salt concentration. In fact, the presence of Na^+^ ions and Cl^−^ ions reduced the strength of the electric field around charged groups in the proteins causing the saturation of their binding sites, and lowering their interactions. The decrease of enthalpy and binding affinity with increasing NaCl (5 to 60 Mm) were also reported for β-conglycinin–lysozyme system [35].

The ITC experiments revealed that βLG and LF complexation was enthalpically favorable and that a small amount of salt ions promoted the interactions between the two proteins. Those results provided one more proof of the major role played by the ionic strength on the complex coacervation of βLG and LF. Understanding the kinetics of the formation of coacervates via a desalting technique is the next step of this study.

### 3.2. LF–βLG Complex Coacervation via Desalting

Unlike the direct mixing presented in Section 3.1.1, desalting is an alternative protocol to better determining the ionic strength sensitivity of a complex coacervation process. Desalting was proposed as a gentle method to build and better control assemblies of nanoparticles and complex biological assemblies [43]. The desalting protocol involves, first, the mixing of the two macromolecules at a sufficiently high ionic strength in which the interactions are inhibited (charge screening). This mixed ‘‘inactive’’ solution is then dialyzed to decrease continuously the ionic strength around the mixed macromolecules [44,45].

In the current study, the behavior during continuous dialysis against MES buffer of mixtures of LF and βLG prepared at three high salt concentrations prior to mixing them has been monitored. The decrease of salt concentration inside the dialysis tubes monitored by conductivity measurements followed an expected exponential behavior (Figure 4). The higher the initial concentration, the longer the dialysis time to reach the final equilibrium concentration. Whatever the initial salt concentration in the range 100 to 400 mM, almost total elimination of salt is achieved after 3.5 h of dialysis in our experimental conditions. Costalat et al. [44] reported the similar kinetics of chloride elimination where 100% of salt was removed after 5 h and 6 h for NaCl starting concentrations of 2 M and 6 M, respectively.

The visual aspect and the corresponding microscopic images of the mixture in the dialysis tube at 100 mM NaCl during 24 h of dialysis are shown in Figure 5. At the beginning of the dialysis, the solution in the tube was almost transparent and no interaction was detected between the two proteins at this ionic strength. Once the dialysis began, the solution in the dialysis tube started getting turbid until reaching a maximum. At the end of the 24 h of dialysis, the solution was once again transparent as the LLPS took place (Figure 5C). This move toward a transparent solution is explained by the progressive formation of highly turbid micro-droplets (microphase separation) that progressively coalesce leading to the observed LLPS.

Turbidity measurements over time during dialysis confirmed the visual and microscopic observations as shown in Figure 6. First, it is worth mentioning that, without added salt, the protein solution reached spontaneously and quickly the maximum value of turbidity as shown for direct mixing experiments (Figure 1). The turbidity decreased continuously in parallel with the occurrence of LLPS inside the dialysis tube. On the other hand, for LF–βLG mixture exposed to high ionic strengths, the turbidity during the dialysis first increased progressively until reaching a maximum and then decreased to almost zero. The areas under the curves are substantially identical for the three-dialysis experiments. The dialysis time needed for maximum turbidity depended on the initial salt concentration, which is to be correlated to the desalting kinetics shown in Figure 4. The ANOVA of the effect of initial NaCl concentration on the maximum of turbidity was significant (*p* < 0.04). The maximum turbidity values at 0 mM and 400 mM were significantly different according to the Tukey’s multiple comparison tests (*p* < 0.05). As illustrated in Figure 4, the highest turbidity value was reached when the salt concentration inside the tube was around 10 mM for the three desalting experiments. The transition from unassociated state to complexation–coacervation between LF and βLG occurred at an optimal and constant ionic strength of around 10 mM, confirming the high salt sensitivity of the studied heteroprotein system. By superimposing the results of Figure 4 and Figure 6, a significant level of turbidity (complex coacervation) was reached around 10 mM NaCl inside the dialysis tubes.

The overall size of the coacervates droplets formed during dialysis experiments varied from 1 to 10 µm as assessed by microscopic observations. However, their number was significantly lower for the sample with an initial NaCl concentration of 400 mM; at the maximum of turbidity, the number of coacervates was equal to 11 ± 3 and 6 ± 2 for 100 mM and 400 mM, respectively (measurements were conducted for images covering a field of 88 × 66 µm). Fresnais et al. [43] reported that dialysis leads to the formation of nanoparticle–polymer clusters with a size three- to fivefold larger than that obtained with direct mixing protocol at the same ionic strength. The same research group concluded that the decrease of the desalting rate led to an increase in the hydrodynamic diameter of copolymers complexes [45]. This explication can fit with the results of the LF–βLG system, as the decrease of the desalting rate is equivalent to conducting dialysis with a higher initial concentration. A higher initial salt concentration delays the on-set of the complexation–coacervation process during dialysis experiments (i.e., turbidity change, Figure 6).

The evolution of the coacervate yield was also monitored during desalting experiments (Figure 7). During the initial dialysis time, like turbidity, the increase of the coacervate yield was slowed down by a higher salt concentration. It is obvious that, without added salt, the coacervate yield sharply increased to a plateau value after 3 h of dialysis. However, in the presence of high salt concentrations, the coacervate yield increased slowly during time owing to the strengthening of the electrostatic interactions and the release of the binding sites that caused the coacervates to form progressively.

Initial high salt concentrations did not affect the final coacervate yield recovered after desalting, which varied from 55% to 65% as statistically assessed by a non-significant ANOVA test (*p* = 0.6) (Figure 7).

## 4. Conclusions

In a previous work, the ability of lactoferrin and β-lactoglobulin to form heteroprotein complex coacervates under optimal conditions of pH and molar stoichiometry of 5.5 and 10, respectively, has been reported in a previous work. Here, the influence of ionic strength on such complex coacervation process was demonstrated using direct mixing and desalting protocols. We showed that, whatever the protocol used, the interaction and subsequent assembly of LF and βLG are highly sensitive to ionic strength of the medium. Compared to the effect of salt on other published systems, the heroprotein complex coacervation between βLG and LF seems to be one of the most sensitive to ionic strength; the molecular interaction between the two proteins can be tuned at a very narrow window of added salt concentration (i.e., 0–5 mM). A concentration of about 20 mM of added NaCl was enough to abolish LLPS between the two proteins, while molecular interactions are still detected at this ionic strength, as proved by ITC experiments. Therefore, the first steps of the coacervation, i.e., the interactions leading to the formation of primary units and soluble complexes, were more resistant to ionic strength than the final steps, i.e., the formation of turbid micrometric droplets. Further research using more sensitive techniques should be conducted in order to identify the salt effect on the soluble complexes and its influence on the structure of the coacervates network. These results are useful for targeting potential applications of LF–βLG complex coacervates as efficient microcapsules for bioactives and texturing agents for food products.

## Figures and Tables

**Figure 1 foods-12-01040-f001:**
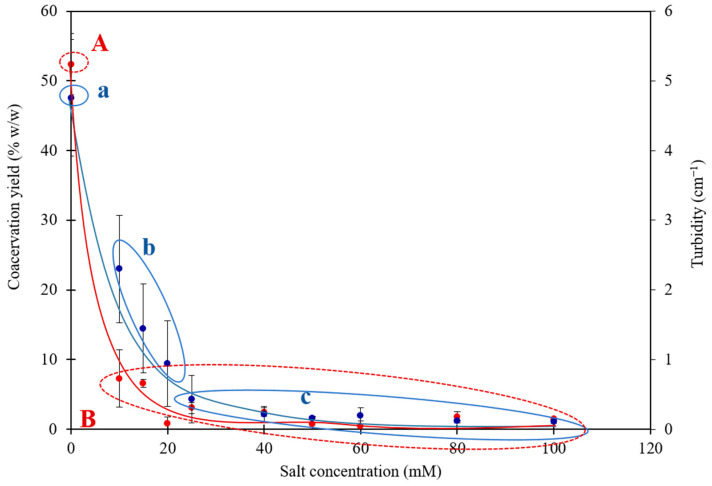
Evolution of lactoferrin–β-lactoglobulin coacervate yield (**red**) and turbidity (**blue**) as a function of salt concentration after direct mixing in 10 mM MES buffer, pH 5.5. Total protein concentration of 0.55 mM. The ANOVA on NaCl concentration was significant (*p* < 4 × 10^−9^ and *p* < 3 × 10^−5^ for the coacervates yield and turbidity, respectively). [a,b,c] and [A,B]: values not sharing the same letter are significantly different according to the Tukey’s multiple comparison test (*p* < 0.05).

**Figure 2 foods-12-01040-f002:**
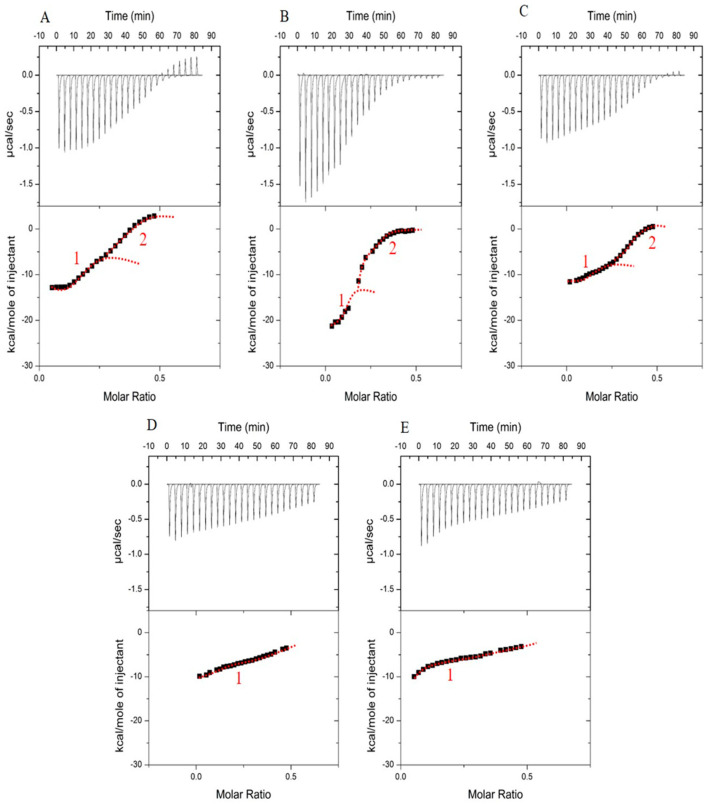
Heat flow thermograms as a function of time (**upper panel**) obtained during the titration of βLG (0.1 mM) by LF (0.25 mM) at different salt concentrations in 10 mM MES buffer pH 5.5 and at 25 °C. **Bottom panel**: graphical representation of the integrated data of enthalpy versus the molar ratio of LF: βLG. (**A**): without added salt; (**B**): with 2.5 mM added NaCl; (**C**): with 5 mM added NaCl; (**D**): with 15 mM added NaCl; (**E**): with 20 mM added NaCl. The red line is just to guide the eyes to distinguish when applicable the two inflection points. Inflection points are indicated by numbers 1 and 2.

**Figure 3 foods-12-01040-f003:**
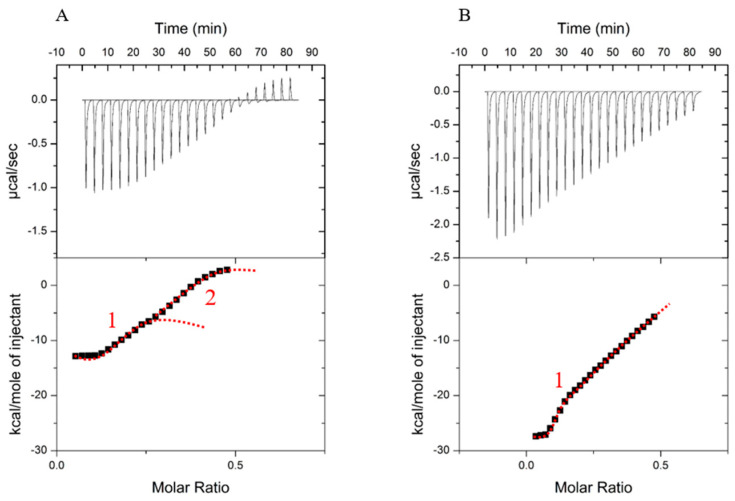
Heat flow thermogram as a function of the time (**upper panel**) obtained during the titration of βLG (0.1 mM) by LF (0.25 mM) in 10 mM MES buffer pH 5.5 at 25 °C (**A**) and 35 °C (**B**). **Bottom panel**: corresponding graphical representation of the integrated data of enthalpy versus the molar ratio of LF:βLG with no added salt. The red line is just to guide the eyes to distinguish when applicable the two inflection points. Inflection points are indicated by numbers 1 and 2.

**Figure 4 foods-12-01040-f004:**
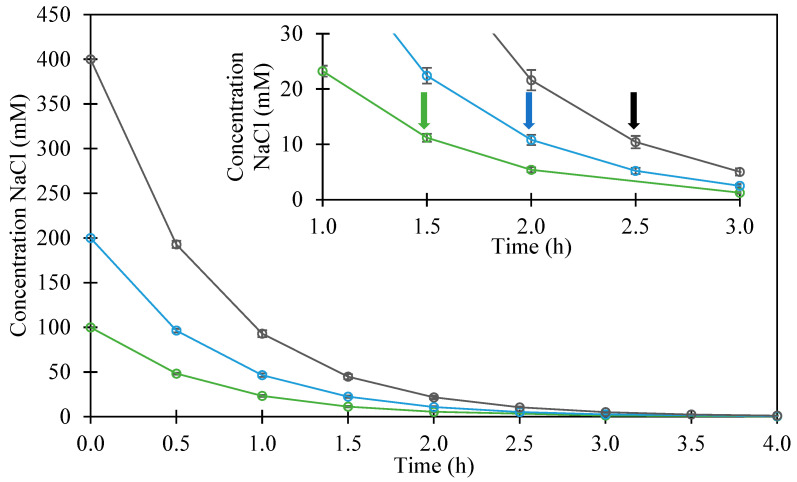
Evolution of ionic strength inside the dialysis tube as monitored by conductivity during 4 h of dialysis of lactoferrin–β-lactoglobulin mixed at various initial ionic strength: 100 mM (**green**), 200 mM (**blue**) and 400 mM (**black**). Insert: zoom on the three first hours. Arrows indicate the time of the maximum of turbidity for each experiment. Dialysis experiments were conducted against 10 mM MES buffer at pH 5.5.

**Figure 5 foods-12-01040-f005:**
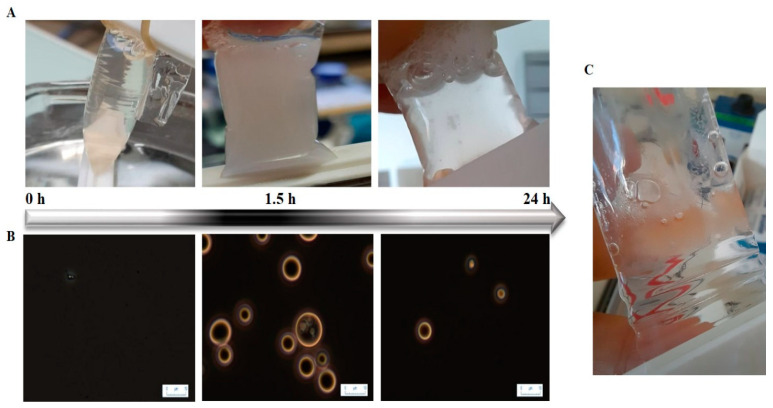
Appearance of lactoferrin–β-lactoglobulin coacervates during desalting protocol as a function of dialysis time for the initial salt concentration of 100 mM. (**A**): visual aspect inside the dialysis tube; (**B**): corresponding microscopic images showing the formation of coacervate droplets; (**C**): an image of the coacervates at the bottom of the dialysis tube at the end of the dialysis experiment. Dialysis was performed against 10 mM MES buffer at pH 5.5. Scale bar: 10 µm.

**Figure 6 foods-12-01040-f006:**
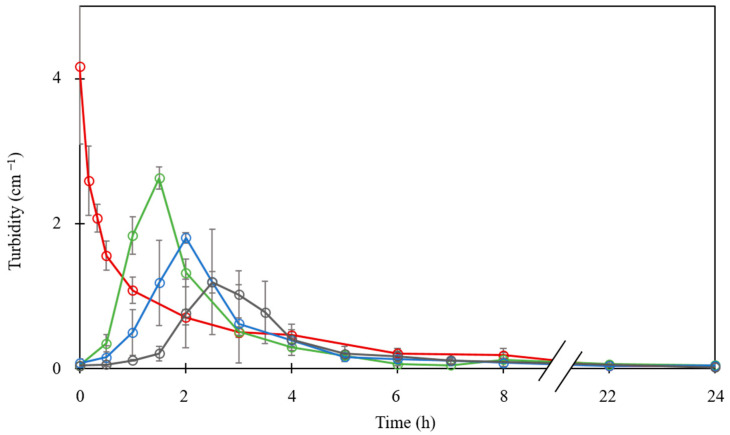
Evolution of turbidity inside the dialysis tube during 24 h desalting experiment of lactoferrin–β-lactoglobulin mixed at total protein concentration of 0.55 mM in 10 mM MES buffer, pH 5.5 at various initial salt concentrations. No added salt (**red**); with added NaCl at concentrations of 100 mM (**green**), 200 mM (**blue**) and 400 mM (**black**).

**Figure 7 foods-12-01040-f007:**
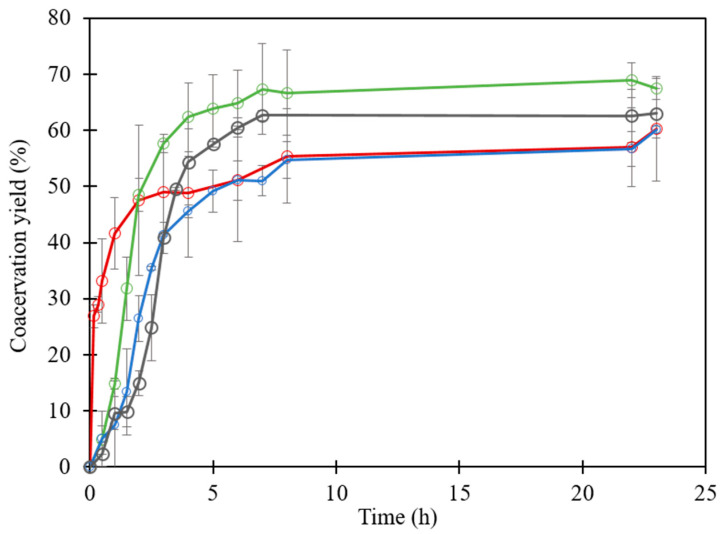
Evolution of the coacervate yield inside the dialysis tube during desalting experiment of lactoferrin–β-lactoglobulin mixed at total protein concentration of 0.55 mM in 10 mM MES buffer, pH 5.5 at various initial salt concentrations. No added salt (**red**); with added NaCl at concentrations of 100 mM (**green**), 200 mM (**blue**) and 400 mM (**black**).

**Table 1 foods-12-01040-t001:** The binding constant (K_a_) and the enthalpy (∆H) as a function of the salt concentration measured [NaCl] by the GUI of PyTC.

[NaCl] (mM)	K_a_ (M^−1^) × 10^5^	∆H (Kcal/mol)
0	4.21 ± 0.06 ^ab^	−18.8 ± 0.04 ^a^
2.5	9.72 ± 0.05 ^ac^	−24.7 ± 0.02 ^c^
5	2.75 ± 0.05 ^ab^	−14.6 ± 0.04 ^ae^
15	0.847 ± 0.02 ^b^	−12.09 ± 0.01 ^bde^
20	0.332 ± 0.015 ^b^	−8.944 ± 0.03 ^d^

a,b,c,d,e: values in a column not sharing the same superscript are significantly different according to the Tukey’s multiple comparison test (*p* < 0.05). The ANOVA on NaCl concentration was significant (*p* < 4 × 10^−5^ and *p* < 0.02 for K_a_ and ∆H, respectively).

## Data Availability

Data are contained within the article.

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
