# Peer review of "Ionic Strength Dependence of the Complex Coacervation between Lactoferrin and β-Lactoglobulin"

_foods, 2023, doi:10.3390/foods12051040_

Round 1
Reviewer 1 Report
In this manuscript entitled “Ionic strength dependence of the complex coacervation between Lactoferrin and β-lactoglobulin” the author described the influence of ionic strength on the complex coacervation between LTF and bLG using direct mixing and desalting protocols. The article is written in good English, and some physicochemical aspects of the coacervation process are discussed. In my opinion, the manuscript requires additional information and explanation.
Major suggestions the authors might want to consider:
- In the introduction, the part should be added discussion about the importance of LTF and BLG in the context of the application of concertation in the food industry.
- Please add the SDS PAGE gel view of control of applied LTF and bLG. In my opinion, relying on the protocol provided by the protein manufacturer/supplier is not enough. Purity and saturation with inorganic salts used during the isolation of proteins (LTF and bLG) are of fundamental importance, e.g. in the context of repeatability of the described results.
- In the discussion part please add issues related to isoelectric point changes related to dispersion stability theories and zeta potential changes.
- Authors wrote: “Overall, our results show the high sensitivity of LF/βLG complex coacervation to salt with a critical NaCl concentration of 20 mM above which the coacervation can no longer occur”. Why? Please extend the discussion.
- Authors postulated: “We assume that this second event is hampered by the reinforcement of the hydrophobic interactions since increasing the ionic strength can enhance those types of interactions”. Please explain the relation between the dominance of hydrophobic and electrostatic interaction.
- Regarding the application of MES buffers. Did the Authors include the interaction between molecules of MES, sodium ions and protein particles?
Based on the above review, the manuscript can be recommended for publication after major revision.
Author Response
Attachment document

Reviewer 2 Report
Comments to the Author
The authors have investigated the influence of ionic strength on the complex coacervation between lactoferrin and β-lactoglobulin using direct mixing and desalting protocols. This is an interesting work that I would suggest reconsidering after the major revisions. The main comments are as follows:
1. Line 27: Please use the acronym LLPS instead of liquid−liquid phase separation.
2. Lines 29-31: A comma seems to be missing after the word “separation”.
3. Line 116: Please add ref.(s) for section 2.3.
4. Lines 185-186: Please add ref.(s) for this sentence.
5. Lines 193-194: It is not suitable to compare with protein/ polysaccharide system.
6. Lines 194-196: Rewrite the sentence.
7. Lines 203-204: If this is a description of the result, add the corresponding graph or table.
8. Lines 262-281: These sentences are repeated.
9. Lines 386-389: Please check the sentence.
10. Line 423: The conclusion needs to be more concise.
11. Line 447: There are too few references in the last three years.
Reviewer 3 Report
General comments: The authors studied the effect of ionic strength on coacervation between lactoferrin and β-lactoglobulin. The study design and methodology were appropriate. The authors explored the mechanism under coacervation, which is important for its application. One major limitation of the study is that the authors did not run (or report) statistical analysis. Such information should be provided in the revision. Additional comments, queries, and suggestions can be found in my specific comments.
Specific comments
Line 9: No need to capitalize “Lactoferrin”. Make changes throughout the manuscript.
Line 48: Change “was” to “were”.
Lines 68-69: Provide the purification method or a reference.
Line 88: Provide a reference.
Line 93: “10 mM MES buffer”.
Line 98: “concentrations”.
Lines 138: In Equation (2), “proteine” should be “protein”.
Line 170: Which value became close to zero at 20 mM?
Line 181: Grammatical error.
Line 187: “subtitle”?
Line 230: Change “were” to “was”.
Line 249: Provide a reference.
Line 257: “10 mM MES buffer”.
Line 284: “10 mM MES buffer”.
Line 293: “studied”.
Line 331: Delete “Zheng et al.”.
Figure 4: Display the error bars in the figure.
Line 388: Delete the first “confirming”.
Round 2
Reviewer 2 Report
The manuscript meets the requirements for publication.
Reviewer 3 Report
The authors have addressed most of my comments. However, using t-test as the statistical analysis method is not acceptable. There are many test conditions such as different salt concentrations and different times, which will result in a large number of pairwise comparisons and a high familywise error rate using t-test. The authors should re-analyze their results using ANOVA followed by a post hoc test that corrects familywise error rate (e.g., Tukey).
